# Anti-Inflammatory Effect and Signaling Mechanism of *Glycine max* Hydrolyzed with Enzymes from *Bacillus velezensis* KMU01 in a Dextran-Sulfate-Sodium-Induced Colitis Mouse Model

**DOI:** 10.3390/nu15133029

**Published:** 2023-07-04

**Authors:** Seung-Hyeon Lee, Ha-Rim Kim, Eun-Mi Noh, Jae Young Park, Mi-Sun Kwak, Ye-Jin Jung, Hee-Jong Yang, Myeong Seon Ryu, Hyang-Yim Seo, Hansu Jang, Seon-Young Kim, Mi Hee Park

**Affiliations:** 1Jeonju AgroBio-Materials Institute, Wonjangdong-gil 111-27, Deokjin-gu, Jeonju-si 54810, Jeollabuk-do, Republic of Korea; sh94@jami.re.kr (S.-H.L.); poshrim@jami.re.kr (H.-R.K.); jingle1234@hanmail.net (E.-M.N.); jjay1205@jami.re.kr (J.Y.P.); 2Kookmin Bio Co., Ltd., 303, Cheonjam-ro, Wansan-gu, Jeonju-si 55069, Jeollabuk-do, Republic of Korea; mskwak@kmbio.co.kr (M.-S.K.); jeongyj@kmbio.co.kr (Y.-J.J.); 3Microbial Institute for Fermentation Industry, Minsokmaeul-gil 61-27, Sunchang 56048, Jeollabuk-do, Republic of Korea; godfiltss@naver.com (H.-J.Y.); rms6223@naver.com (M.S.R.); 4Jeonbuk Institute for Food-Bioindustry, Wonjangdong-gil 111-18, Deokjin-gu, Jeonju-si 54810, Jeollabuk-do, Republic of Korea; hiseo@jif.re.kr (H.-Y.S.); jhs@jif.re.kr (H.J.)

**Keywords:** enzymatic hydrolysis, *Glycine max*, *Bacillus velezensis* KMU01, inflammation, dextran sulfate sodium, colitis

## Abstract

The purpose of this study was to investigate the effect that *Glycine max* hydrolyzed with enzymes from *Bacillus velezensis* KMU01 has on dextran-sulfate-sodium (DSS)-induced colitis in mice. Hydrolysis improves functional health through the bioconversion of raw materials and increase in intestinal absorption rate and antioxidants. Therefore, *G. max* was hydrolyzed in this study using a food-derived microorganism, and its anti-inflammatory effect was observed. Enzymatically hydrolyzed *G. max* (EHG) was orally administered once daily for four weeks before DSS treatment. Colitis was induced in mice through the consumption of 5% (*w*/*v*) DSS in drinking water for eight days. The results showed that EHG treatment significantly alleviated DSS-induced body weight loss and decreased the disease activity index and colon length. In addition, EHG markedly reduced tumor necrosis factor-α, interleukin (IL)-1β, and IL-6 production, and increased that of IL-10. EHG improved DSS-induced histological changes and intestinal epithelial barrier integrity in mice. Moreover, we found that the abundance of 15 microorganisms changed significantly; that of Proteobacteria and *Escherichia coli*, which are upregulated in patients with Crohn’s disease and ulcerative colitis, decreased after EHG treatment. These results suggest that EHG has a protective effect against DSS-induced colitis and is a potential candidate for colitis treatment.

## 1. Introduction

Bioactive peptides can be generated from dietary proteins through fermentation and enzymatic hydrolysis techniques. These peptides hold the potential for utilization in the pharmaceutical and food industries as nutraceuticals, offering health-enhancing properties [1]. Intact and hydrolyzed soy proteins are commonly recognized as primary nitrogen sources in infant and adult formulas [2]. Soy proteins exert various physiological functions, such as reducing cholesterol levels and body fat [3]. Several antihypertensive peptides have been reported from enzymatically hydrolyzed soybean proteins and fermented soybean products; therefore, bioactive soybean peptides have a wide range of health benefits [4,5,6]. Food-derived, physiologically active peptides have received considerable attention owing to their applications and efficacy in the prevention and treatment of various chronic diseases [7]. Shorter peptides are more easily absorbed compared with longer ones due to their ability to efficiently traverse the intestinal barrier and exert cellular-level effects [8]. Soybeans (*Glycine max*) have biological functions that are effective in the treatment of atherosclerosis, inflammatory bowel disease, and several cancers [9,10]. Soybeans offer a rich source of nutritious compounds and enjoy widespread acceptance among the public. Moreover, they have received FDA approval for their potential to reduce the risk of chronic ailments, including coronary heart disease [7]. In addition, active peptides from soybeans have been reported to be effective at treating various diseases [6]. The soybean-derived tripeptide, Leu-Ser-Trp (LSW), ameliorates vascular endothelial oxidative stress and inflammatory responses [11]. In a recent study, it was observed that traditional Korean fermented soybean foods, such as doenjang, demonstrated protective effects against colitis induced by dextran sulfate sodium (DSS) in a mouse model [12]. Studies have indicated that Cheonggukjang, a type of fermented soybean, may offer protection against colitis-associated colorectal cancer in mice [13].

Inflammatory bowel disease (IBD) is a chronic condition characterized by inflammation, with Crohn’s disease and ulcerative colitis (UC) being the two primary forms. Common symptoms of IBD include weight loss, abdominal pain, diarrhea, and rectal bleeding [14]. While anti-inflammatory drugs are employed for the treatment of UC, their clinical application is hindered by the presence of various side effects [15]. Currently, there are no successful therapies for managing IBD. On the other hand, natural products have demonstrated effectiveness in both experimental models and clinical trials for the treatment of IBD. They have shown the ability to preserve the integrity of the intestinal epithelial barrier, regulate macrophage activation, modulate innate and adaptive immune responses, and inhibit the activity of tumor necrosis factor (TNF)-α [16]. The objective of this study was to explore natural products and their underlying mechanisms of action for the treatment of IBD. To assess drug effectiveness and toxicity, animal models of colitis have been employed in preclinical trials [17]. The utilization of a mouse model of colitis induced by DSS, a sulfated polysaccharide with diverse molecular weights, offers several advantages, including simplicity, speed, control, and reproducibility [18]. The mouse model of colitis induced by DSS closely resembles human IBD, exhibiting characteristic symptoms including weight loss, diarrhea, and the presence of blood in stools [5]. In the context of DSS-induced colitis, there is an observed increase in the intercellular distance between mucosal cells within the crypts and vascular endothelial cells [19]. DSS additionally impacts the mucosal barrier and exhibits toxicity towards cells in the intestinal epithelium located beneath it [20]. Therefore, a DSS-induced colitis mouse model is essential for inflammatory colitis research. We used this mouse model to demonstrate the effects of enzymatically hydrolyzed *G. max* (EHG) using food-derived microorganisms.

Here, we found that soybeans fermented using the food-derived microorganism, *Bacillus velezensis* strain KMU01, which has γ-glutamyl transferase activity [9], alleviated DSS-induced colitis. Therefore, we investigated the anti-inflammatory effects and molecular mechanisms of fermented soybeans, including various peptides, using a DSS-induced colitis mouse model and evaluated their potential for the treatment of IBD. 

## 2. Materials and Methods

### 2.1. Preparation of Soybean Enzymatic Hydrolysate

The *B. velezensis* KMU01 strain isolated from kimchi, a fermented Korean food, was cultured in a medium optimized for protease production at 37 °C for 12 h. The filtrate (from which the cells were removed through centrifugation and filtration) was mixed with soybean (*G. max*) powder (Jeollabuk-do, Korea) at 40 °C for 4 h. Thereafter, the samples were freeze-dried before use in subsequent experiments.

### 2.2. Animals

Male BALB/c mice (specific pathogen-free grade) were obtained from Damul Science (Daejeon, Korea) at five weeks of age and allowed to acclimate for one week. The mice were housed in a controlled environment with a 12 h light/dark cycle, maintained at a temperature of 22 ± 2 °C, and a relative humidity of 55 ± 5%.

Specific pathogen-free-grade BALB/c mice (male, five-weeks-old) were purchased from Damul Science (Daejeon, Korea) and acclimated for one week. The mice were kept in a room with a 12 h light/dark cycle, at a temperature of 22 ± 2 °C and a relative humidity of 55 ± 5%. The Animal Care Committee of the Jeonju AgroBio-Materials Institute provided approval for all experimental procedures conducted in this study (JAMI IACUC 2022006, Jeonju, Korea).

### 2.3. Experimental Groups

The animals were divided into four treatment groups (seven mice/group): N (normal), NC (negative control; 5% DSS), PC (positive control; 5% DSS + 50 mg/kg/day 5-aminosalicylic acid [5-ASA]), and EHG (5% DSS + 1 g/kg/day EHG). The N and NC groups were administered distilled water. Before DSS therapy, the PC and sample groups received a once daily oral administration for four weeks.

### 2.4. Induction of Colonic Inflammation

Using DSS, acute colitis was induced. For eight consecutive days, the mice were given access to drinking water containing 5% (*w*/*v*) DSS (36–50 kDa), and one day later, they were euthanized. Before administering the samples each day following DSS treatment, the body weights and disease activity index (DAI) were measured. The ascribed DAI score depended on the features of the stool and varied from 0 (normal) to 4 (maximal disease activity) as previously described [12] (Table 1).

### 2.5. Evaluation of Biomarkers in Serum and Colon Tissue 

ELISA kits (R&D Systems, Abingdon, UK) were utilized to measure the serum levels of TNF-α, interleukin (IL)-1β, IL-6, and IL-10, following the manufacturer’s instructions. In ice-cold RIPA buffer (Invitrogen, Carlsbad, CA, USA) with a protease and phosphatase inhibitor cocktail (Thermo Fisher Scientific, Waltham, MA, USA), the colon tissues were homogenized. SDS-PAGE was used to separate the protein samples (25 μg per lane), which were then transferred to polyvinylidene difluoride membranes (Bio-Rad, Munich, Germany). The blots were examined with the designated antibodies and produced using an enhanced chemiluminescence (ECL) system (Amersham, Buckinghamshire, UK). Densitometric scanning (Amersham Imager 600; GE Healthcare, Buckinghamshire, UK) was used for the quantitative analysis. Quantitative real-time reverse transcription PCR (qRT-PCR) was used to measure the mRNA expression levels of inflammatory cytokines. Colon tissues were homogenized using Hybrid-RTM (GeneAll, Seoul, Korea). A cDNA Synthesis Kit (BioFact, Daejeon, Korea) was then used to reverse transcribe 1 μg of RNA in order to create cDNA. The 2× RT Pre-Mix (BioFact) was used to carry out the qRT-PCR. Table 2 provides a list of the used primer sequences. The comparative Ct method was used to determine the relative mRNA levels, with β-actin serving as the reference gene. 

### 2.6. Histology

Mouse colon tissues were embedded in paraffin after being treated in 4% paraformaldehyde. Slices of the tissue sections, 4 μm thick, were made and stained with either alcian blue (AB) or hematoxylin and eosin (H&E). The sections were deparaffinized, rehydrated, incubated with antibodies overnight at 4 °C, and then treated with an anti-rabbit Envision Plus Polymer Kit (Dako, Glostrup, Denmark) for the immunohistochemistry (IHC) examination of ZO-1 and occludin. The sections were subjected to hematoxylin staining. Using an optical microscope (Olympus, Tokyo, Japan), morphological characteristics of the stained sections and IHC expression were studied and photographed on camera. Using the average of the grades for each criterion, the histological scores were calculated.

### 2.7. Cell Culture and Viability

RAW264.7 cells were grown in Dulbecco’s Modified Eagle Medium (Gibco, Waltham, MA, USA), which also contains 1% penicillin (100 U/mL)/streptomycin (Gibco) and 10% heat-inactivated fetal bovine serum (Gibco). The cell cultures were kept at a temperature of 37 °C and 5% CO₂. After being seeded in a 96-well plate at a density of 5 × 10⁵ cells/mL, the cells were exposed to various doses of EHG (1, 10, 100, 250, 500, and 1000 μg/mL) for 24 h to determine their viability. The CellTiter Cell Proliferation Test Kit (Promega, Madison, WI, USA) was used for the MTS assay, and the analysis was carried out in accordance with the manufacturer’s instructions.

### 2.8. Evaluation of Biomarkers in RAW264.7 Cells

Prior to receiving 1 μg/mL LPS treatment for 24 h, the RAW264.7 cells were pretreated with EHG (100, 250, and 500 μg/mL) for 1 h. The cells were used for Western blotting, and the supernatant was collected and used to measure the nitric oxide (NO) and inflammatory cytokines. Griess reagent (Promega) was used to detect NO, and ELISA kits (R&D Systems) were used to detect TNF-α, IL-1β, IL-6, and IL-10. The results were analyzed in accordance with the manufacturer’s instructions. Protease and phosphatase inhibitor cocktails (Thermo Fisher Scientific) were added to ice-cold RIPA buffer (Invitrogen) to lyse the cells. SDS-PAGE was used to separate protein samples (20 μg per lane), which were then transferred to polyvinylidene difluoride membranes (Bio-Rad). The blots were examined with the designated antibodies and produced using an enhanced chemiluminescence (ECL) system (Amersham). Densitometric scanning (Amersham Imager 600; GE Healthcare) was used for the quantitative analysis.

### 2.9. Gut Microbiome Analysis Using Next-Generation Sequencing

For each fecal sample, the 16S rRNA sequences were processed using Mothur v.1.36, in accordance with the MiSeq standard operating procedures. Using Silva Reference Alignment v.12350 (Bremen, Germany) to align the sequences, bacterial counts and taxonomy identification were carried out as previously mentioned [21]. Each sample’s relative bacterial counts were determined using taxonomic groupings, and the fecal bacteria’s α- and β-diversity were evaluated. An unweighted UniFraq distance matrix was used to generate the β-diversity, which was then displayed using the R program. Using permutation-based variance analysis, the separation between each group was examined. Through network analysis, the relationships between the gut bacteria at the genus level, as well as those between visceral fat mass, SCFA, and glucose metabolism, were identified.

### 2.10. Statistical Analysis

Sigmaplot v16.0 software (Systat Software Inc., San Jose, CA, USA) was used for all of the statistical analyses. Data are expressed as the means ± standard deviation (SD). To identify differences, a statistical analysis was conducted, followed by a one-way analysis of variance and a Duncan’s multiple comparison test. These analyses were used to evaluate the differences among three or more groups for all of the measured parameters. Statistical significance was set at *p* < 0.05.

## 3. Results

### 3.1. Effect of EHG on DSS-Induced Colitis

Previous studies have documented that mice induced with DSS exhibit disease progression characterized by weight loss, colon shortening, and the presence of bloody stools. Following a four-week period of oral administration of hydrolyzed soybean, the mice were provided with drinking water containing 5% DSS for eight days to induce UC. During the four-week treatment period, body weight changes were not observed after treatment with EHG (Figure 1A). In the DSS-treated NC group, a significant reduction in body weight and colon length was observed, accompanied by an increase in DAI compared with that of the N group. The administration of EHG, in contrast, demonstrated significant effectiveness at preventing DSS-induced weight loss and the increase in DAI when compared with the NC treatment (Figure 1B–D). The groups treated with EHG exhibited a significant inhibition of the DSS-induced decrease in colon length compared with the NC group (Figure 1E,F). Hence, it can be inferred that EHG possesses therapeutic potential for DSS-induced UC, similar to that of 5-ASA.

### 3.2. Effects of EHG on Colonic Histological Damage and Mucin in Mice with DSS-Induced Colitis

Histological examination of the colon tissues was performed using H&E and AB staining to evaluate the observed changes. In the DSS-treated group, notable histopathological alterations were observed, including inflammatory cell infiltration, loss of goblet cells, crypt deterioration, and submucosal edema, as compared with the N group. On the other hand, treatment with EHG effectively mitigated DSS-induced intestinal damage and inflammation in the colon (Figure 2A,C). AB staining was employed to detect the presence of mucin in the intestinal goblet cells of the colon. Abundant mucin was observed in the colon tissues of the control group; however, a significant decrease in mucin content was noted in the DSS-treated group. In contrast, the administration of EHG resulted in a significant increase in the amount of mucin present in the colon (Figure 2B,D).

### 3.3. Effect of EHG on DSS-Induced Pro-Inflammatory Cytokine Production

The levels of TNF-α, IL-6, and IL-1β were measured in both the serum and colon tissue using ELISA and RT-qPCR. DSS significantly increased the levels of TNF- α, IL-6, and IL-1β; however, that of IL-10 decreased. After the administration of EHG for approximately four weeks, the production of TNF-α (Figure 3A,E), IL-6 (Figure 3B,F), and IL-1β (Figure 3C,G) decreased, whereas that of IL-10 increased (Figure 3D,H).

### 3.4. Effects of EHG on iNOS and COX-2 Expression and the NF-κB Signaling Pathway in Mice with DSS-Induced Colitis

In the context of DSS-UC, there is a significant upregulation of iNOS and COX-2 expression. iNOS is known to mediate immune responses through the production of NO, while COX-2 is induced by pro-inflammatory cytokines. Compared with the NC group, the DSS-treated group exhibited a notable increase in the protein expression of COX-2 and iNOS. In contrast, mice treated with EHG demonstrated a significant reduction in the expression of COX-2 and iNOS compared with the DSS-induced control group (Figure 4). 

We found that DSS treatment resulted in the activation of NF-κB phosphorylation in colonic tissue. However, the administration of EHG significantly attenuated this effect, indicating its potential as a modulator of NF-κB signaling in the context of DSS-induced colitis. Notably, EHG exhibited remarkable efficacy in inhibiting DSS-induced NF-κB phosphorylation, mirroring its inhibitory effect on COX-2 expression. These findings suggest that EHG may exert its protective effects against DSS-induced colitis, at least in part, through the modulation of NF-κB signaling pathways.

### 3.5. Effects of EHG on Intestinal Barrier Function in Mice with DSS-Induced Colitis

To evaluate the integrity of the intestinal epithelial barrier, we performed immunohistochemistry (IHC) to assess the expression of tight junction (TJ) proteins, specifically ZO-1 and occludin. ZO-1 and occludin exhibited uniform expression along the superficial membrane, as well as within the crypts of the epithelium in the N group. On the other hand, the DSS-treated group showed compromised integrity of the colonic barrier and disrupted crypt structure, leading to a decrease in the expression of TJ proteins. In contrast, the administration of EHG effectively preserved and maintained the integrity of the intestinal barrier, as evidenced by the high expression levels of ZO-1 and occludin along the epithelial membranes and the presence of intact colonic crypts (Figure 5). The results suggest that the administration of EHG provides protection against the disruption of intestinal barrier function induced by DSS, as it effectively prevents the loss of TJ proteins.

### 3.6. Effect of EHG on Microbiome Diversity in Mice with DSS-Induced Colitis

The 15 microorganisms that had the greatest effect on the differences in microbial community structure between each group after the ingestion of the test product are shown in the biomarker discovery results in Table 3. Interestingly, in the N group, Clostridiales, which is related to immune enhancement, accounted for a high percentage, and in the DSS-treated group, its abundance was lower; however, this effect was reversed after the administration of EHG. In addition, similar to the results of studies confirming an increase in Proteobacteria and *Escherichia coli* in patients with Crohn’s disease and UC, we confirmed that both were increased during disease induction, and the ratio of the corresponding microorganisms decreased after treatment with EHG.

### 3.7. Anti-Inflammatory Effect of EHG on LPS-Induced RAW264.7 Cells

To confirm the anti-inflammatory effects and related mechanisms of EHG at the cellular level, we examined these in RAW264.7 cells after LPS induction. LPS-induced inflammation using mouse macrophage is a widely utilized in vitro model to study inflammatory processes and evaluate the potential anti-inflammatory effects of various compounds [22]. In our initial experiments, we demonstrated that EHG exhibited no cytotoxic effects at concentrations up to 1000 μg/mL (Figure 6A). To evaluate the suppressive properties of EHG on LPS-induced nitric oxide (NO) production, RAW 264.7 cells were exposed to different concentrations of EHG and subsequently incubated with or without LPS (1 μg/mL). Figure 6B indicates that treatment with EHG in RAW 264.7 cells stimulated with LPS resulted in a notable dose-dependent reduction in the production of nitric oxide (NO). In addition, our findings demonstrated that EHG treatment in a dose-dependent manner effectively reduced the secretion of pro-inflammatory cytokines, such as TNF-α, IL-6, and IL-1β, from LPS-treated macrophages (Figure 6C–E). Moreover, EHG treatment significantly increased the release of IL-10 in LPS-induced macrophages compared with the control group treated with LPS alone. Furthermore, our results revealed that EHG treatment effectively inhibited the protein expression of iNOS induced by LPS and simultaneously suppressed the expression of COX-2. Moreover, LPS treatment stimulated phosphorylation of NF-κB in RAW264.7 cells; however, this was significantly decreased in a dose-dependent manner upon treatment with EHG (Figure 6G,H).

Taken together, these findings suggest that the administration of EHG exerts its effects on the inflammatory response in RAW264.7 cells by regulating the NF-κB signaling pathway.

## 4. Discussion

The field of natural product research is experiencing continuous growth, with ongoing efforts to enhance their biological activity for the treatment and prevention of diverse diseases. Novel approaches are actively being explored to maximize the potential of natural products in various therapeutic applications. Several reports have suggested that the hydrolysis of natural products improves functional health through the bioconversion of raw materials and increase in intestinal absorption rate and antioxidants [23]. Therefore, in this study, *G. max* was hydrolyzed using the *B. velezensis* KMU01 strain from traditional Korean fermented kimchi, and its potential therapeutic benefits in mitigating colitis-associated inflammation. Despite extensive research on the pathogenesis of UC and ongoing efforts to identify novel treatment approaches, a definitive cure for the disease has yet to be developed. Medications for IBD often result in possible side effects and there is a risk of complications. The long-term use of drugs commonly prescribed for UC treatment, such as aminosalicylates and corticosteroids, is often restricted due to their significant side effects [24]. Hence, there is a pressing need to develop pharmaceuticals that are both safer and more efficacious in the treatment of UC. Extensive research has been undertaken to investigate the therapeutic potential of natural products and their bioactive compounds as potential remedies for UC [24,25].

Our findings demonstrate that the administration of EHG had beneficial effects on colitis, as evidenced by improvements in parameters such as body weight, colon length, histological features, and DAI scores. The administration of EHG exhibited significant recovery in terms of body weight and colon length. Furthermore, EHG demonstrated a restoration of the contractile force in intestinal tissue, indicating an improvement in intestinal motility. In summary, the findings strongly indicate that EHG exerts a notable preventive effect against UC.

Moreover, our investigation unveiled that EHG effectively mitigated DSS-induced colitis by modulating the expression of multiple pro-inflammatories. Previous studies have highlighted the crucial role of pro-inflammatory cytokines in the development of colitis, emphasizing the significance of anti-cytokine therapies, particularly TNF-specific agents, as a fundamental approach in the clinical management of UC and Crohn’s disease. The administration of TNF-specific antibodies has demonstrated efficacy in suppressing persistent intestinal inflammation and promoting mucosal healing in patients with IBD. These antibodies specifically target tumor necrosis factor (TNF), a key pro-inflammatory cytokine implicated in the pathogenesis of IBD, and their use has become a valuable therapeutic strategy for managing the disease [26], and IL-6 plays a critical role in promoting inflammation by activating various target cells, such as antigen-presenting cells and T cells [4]. IL-10 is a potent anti-inflammatory cytokine that plays a crucial role in regulating immune responses. It exerts its effects by inhibiting antigen presentation and suppressing the release of pro-inflammatory cytokines [27]. Emerging research has provided compelling evidence that loss-of-function mutations in genes encoding IL-10 and IL-10R (IL-10 receptor) are associated with a distinct form of inflammatory bowel disease (IBD) [28]. Our findings demonstrate that treatment with EHG in mice with DSS-induced colitis resulted in a significant modulation of the cytokine levels. Specifically, the pro-inflammatory cytokines TNF-α, IL-6, and IL-1β were found to be decreased, while the anti-inflammatory cytokine IL-10 was increased. These changes in cytokine profile indicate a shift towards a more anti-inflammatory environment in the colonic tissues of DSS-induced colitis mice through the treatment of EGH.

Furthermore, our results indicate that EHG treatment effectively suppressed the expression of iNOS and COX-2 in the colon tissue of mice with DSS-induced colitis. iNOS and COX-2 are key enzymes involved in the production of inflammatory mediators, such as nitric oxide (NO) and prostaglandins, respectively. The upregulation of iNOS and COX-2 is commonly associated with inflammatory conditions and plays a crucial role in the progression of colitis. The ability of EHG to attenuate the expression of these inflammatory enzymes suggests its potential as a therapeutic agent for mitigating colonic inflammation and associated symptoms. Furthermore, our study revealed that EHG treatment exerted inhibitory effects on NF-κB, a crucial transcription factor involved in the pathogenesis of IBD. By targeting NF-κB, EHG may contribute to the suppression of inflammatory mediators and the restoration of immune homeostasis. These findings highlight the therapeutic potential of EHG as a modulator of NF-κB signaling in the context of IBD.

Fermented soybean foods are known to contain a variety of bioactive compounds with potential therapeutic effects. These include low-molecular-weight peptides, melanoidins, furanones, 3-hydroxyanthranilic acid, phenolic components, and antioxidant components. These compounds have been associated with various health benefits, including anti-inflammatory, antioxidant, and antimicrobial properties [22]. Fermented soy foods offer potential therapeutic benefits for hypertension, thanks to bioactive compounds such as angiotensin I-converting enzyme inhibitory peptides and γ-aminobutyric acid (GABA) [29]. Fermented soybean foods contain various potential anti-inflammatory bioactive constituents, including γ-linolenic acid, butyric acid, soy polysaccharides, 2S albumin, and isoflavone glycols. These compounds have demonstrated anti-inflammatory properties and may contribute to the overall health benefits of fermented soy foods. Additionally, certain components such as deoxynojirimycin, genistein, and betaine have a shown significant activity against α-glucosidase, an enzyme involved in carbohydrate metabolism [22]. These findings suggest that fermented soybean foods have the potential to be utilized in the development of functional foods or dietary interventions targeting inflammation and metabolic disorders. In addition to their anti-inflammatory and metabolic benefits, fermented soybean foods are also known to contain neuroprotective components. These include isoflavones, vitamin B12, indole alkaloids, arbutin, and nattokinase. These compounds have been studied for their potential to support brain health and protect against neurodegenerative diseases [30]. The anticancer activity of fermented soybean foods can be attributed to the various bioactive components present in them. These include isoflavones, furanones, trypsin inhibitors, surfactins, and 3-hydroxyanthranilic acid [22]. We also identified various functional peptides in EHG. therefore, further studies are needed to identify its functional peptides and bioactive components. In addition, we found that the abundance of Proteobacteria and *E. coli* increased during disease induction and the ratio decreased after EHG treatment, similar to the results of studies confirming an increase in both in patients with Crohn’s disease and UC. Based on our results regarding the anti-colitis effect of EHG, clinical studies are needed to confirm whether it could serve as an alternative treatment solution.

## 5. Conclusions

The findings from our study provide compelling evidence for the significant anti-colitis effects of G. max hydrolyzed with *B. velezensis* KMU01. These effects were observed through the modulation of the NF-κB signaling pathway and regulation of the intestinal epithelial barrier in mice with DSS-induced colitis. Based on these results, EHG derived from G. max holds great promise as a natural therapeutic agent for the treatment of colitis. The utilization of EHG as a natural source-derived therapy could offer a safer and more effective alternative for individuals suffering from colitis.

## Figures and Tables

**Figure 1 nutrients-15-03029-f001:**
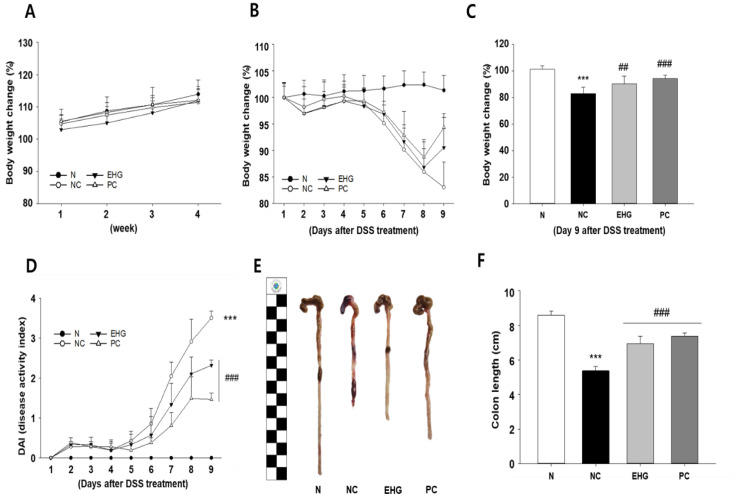
Effects of enzymatically hydrolyzed *Glycine max* (EHG) on body weight, disease activity index (DAI), and colon length in mice with colitis induced by dextran sulfate sodium (DSS). (**A**) Change in body weight after treatment with EHG. (**B**) Change in body weight after the administration of DSS. (**C**) Change in body weight nine days after colitis was induced. (**D**) Disease activity index. (**E**) Colon paragraph and (**F**) length. N (normal), NC (negative control; 5% DSS), EHG (EHG + 5% DSS), and PC (positive control; 5% DSS + 5-aminosalicylic acid (5-ASA)) groups are shown. All values represent the mean ± standard deviation (SD) (*n* = 7/group). The data obtained were subjected to analysis using Duncan’s multiple comparison test. *** *p* < 0.001 vs. N; ## *p* < 0.01 and ### *p* < 0.001 vs. NC.

**Figure 2 nutrients-15-03029-f002:**
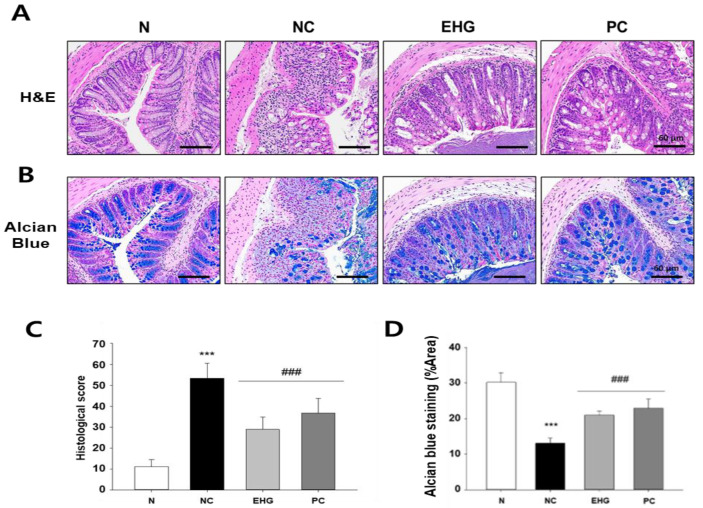
Effect of EHG on histological assessment in mice with DSS-induced colitis. Representative images of colon tissue stained with hematoxylin and eosin (H&E) (**A**) and alcian blue (**B**). Assessment of the (**C**) histological score and (**D**) percent area of alcian blue staining. Magnification: 200×, scale bar: 60 μm. N (normal), NC (negative control; 5% DSS), EHG (EHG + 5% DSS), and PC (positive control; 5% DSS + 5-ASA) groups are shown. All values represent the mean ± SD (*n* = 7/group). The data obtained were subjected to analysis using Duncan’s multiple comparison test. *** *p* < 0.001 vs. N; ### *p* < 0.001 vs. NC.

**Figure 3 nutrients-15-03029-f003:**
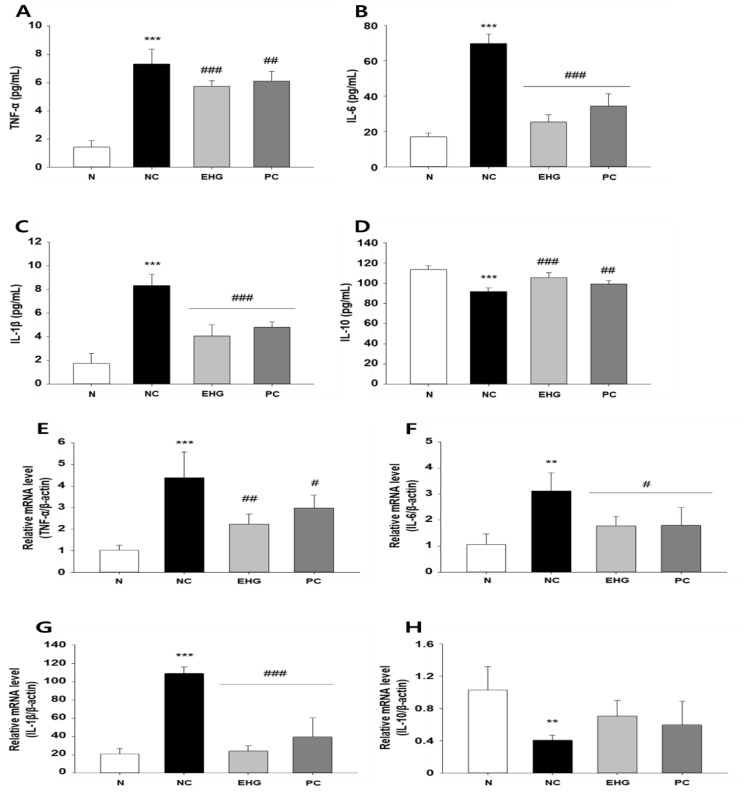
Effects of EHG on pro- and anti-inflammatory cytokines in the serum and colon tissues of mice with DSS-induced colitis. Levels of (**A**) tumor necrosis factor (TNF)-α, (**B**) interleukin (IL)-6, (**C**) IL-1β, and (**D**) IL-10 in DSS-induced colitis mouse serum. Levels of (**E**) TNF-α, (**F**) IL-6, (**G**) IL-1β, and (**H**) IL-10 in DSS-induced colitis mouse colon tissues. N (normal), NC (negative control; 5% DSS), EHG (EHG + 5% DSS), and PC (positive control; 5% DSS + 5-ASA) groups are shown. All values represent the mean ± SD (*n* = 7/group). The data obtained were subjected to analysis using Duncan’s multiple comparison test. ** *p* < 0.01 and *** *p* < 0.001 vs. N; # *p* < 0.05, ## *p* < 0.01, and ### *p* < 0.001 vs. NC.

**Figure 4 nutrients-15-03029-f004:**
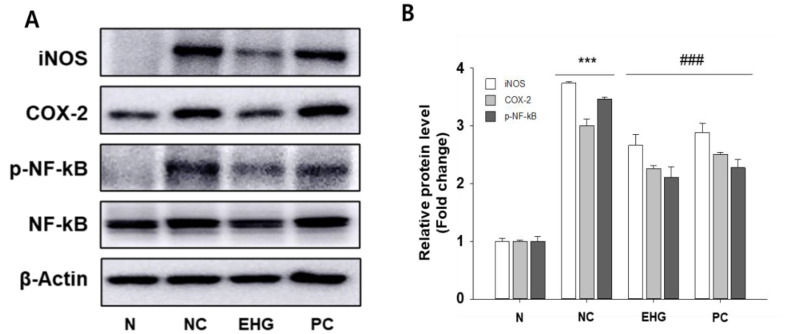
Effect of EHG on the inflammatory signaling pathway in colon tissues of mice with DSS-induced colitis. (**A**) Protein expression levels of iNOS and COX-2, as well as the phosphorylation status of NF-κB, in the colon was assessed using immunoblotting analysis. (**B**) Relative protein level of protein band was quantified by densitometry using image analysis. The density of iNOS and COX-2 was normalized by β-actin and phosphorylated NF-κB normalized to the total NF-κB. N (normal), NC (negative control; 5% DSS), EHG (EHG + 5% DSS), and PC (positive control; 5% DSS + 5-ASA) groups are shown. All values represent the mean ± SD (*n* = 7/group). The data obtained were subjected to analysis using Duncan’s multiple comparison test. *** *p* < 0.001 vs. N; ### *p* < 0.001 vs. NC.

**Figure 5 nutrients-15-03029-f005:**
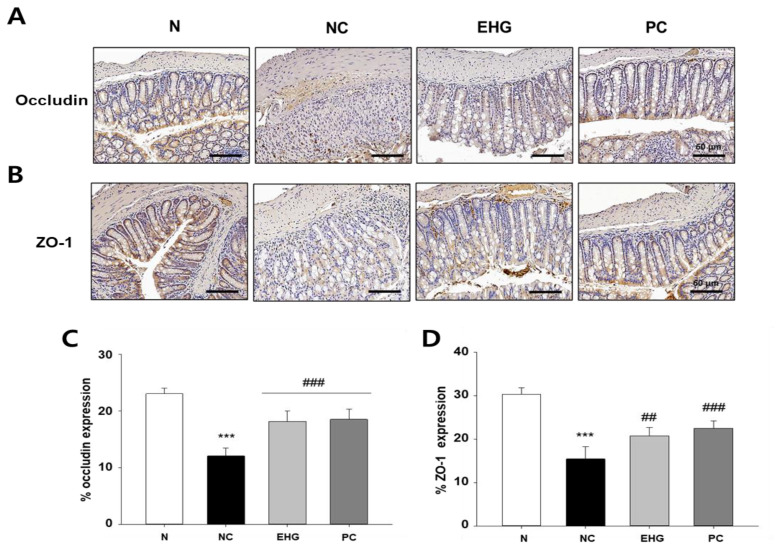
Effects of EHG on occludin and ZO-1 expression in mice with DSS-induced colitis. Immunohistochemical expression patterns of the tight-junction-related proteins, including (**A**) ZO-1 and (**B**) occludin, in colon tissue. Magnification: 200×, scale bar: 60 μm. (**C**,**D**) Histological scoring of occludin and ZO-1 expression. N (normal), NC (negative control; 5% DSS), EHG (EHG + 5% DSS), and PC (positive control; 5% DSS + 5-ASA) groups are shown. All values represent the mean ± SD (*n* = 7/group). The data obtained were subjected to analysis using Duncan’s multiple comparison test. *** *p* < 0.001 vs. N; ## *p* < 0.01 and ### *p* < 0.001 vs. NC.

**Figure 6 nutrients-15-03029-f006:**
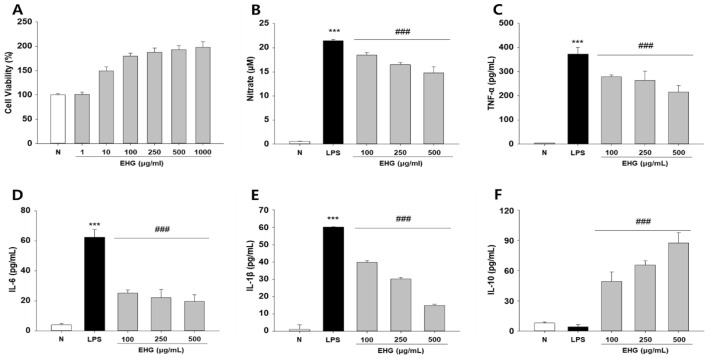
Effects of EHG on LPS-induced nitric oxide (NO) and cytokine production, as well as the NF-ĸB pathway, in RAW264.7 cells. (**A**) Cell viability measurements. (**B**) NO production determined using the Griess reaction in supernatants from RAW264.7 cells. Levels of (**C**) TNF-α, (**D**) IL-6, (**E**) IL-1β, and (**F**) IL-10 in RAW264.7 macrophages. (**G**) Protein expression of iNOS, COX-2, and components of the NF-ĸB signaling pathway were assessed using immunoblotting. Equal amounts of total protein were analyzed via SDS-PAGE and the quantitative analysis of the indicated blots is shown. The densitometry data are expressed as the relative density of protein bands, which were normalized to the level of β-actin (**H**). All values represent the mean ± SD from three independent experiments. The data obtained were subjected to analysis using Duncan’s multiple comparison test. *** *p* < 0.001 vs. N (normal group); # *p* < 0.05, ### *p* < 0.001 vs. LPS.

**Table 1 nutrients-15-03029-t001:** Disease activity index score.

Weight Loss (%)	Shape of Stool	Occult Blood/Bloody Stool	Score
0	Normal	Negative	0
1–5	Soft stool	Negative	1
6–10	Soft stool	Occult blood	2
11–15	Diarrhea	Occult blood	3
>15	Diarrhea	Bloody stool	4

**Table 2 nutrients-15-03029-t002:** Primer sequences used for quantitative real-time reverse transcription PCR.

Gene	Forward (5′-3′)	Reverse (5′-3′)
*TNF-* *α*	AACTAGTGGTGCCAGCCGAT	CTTCACAGAGCAATGACTCC
*IL-6*	TGTCTATACCACTTCACAAGTCGGAG	GCACAACTCTTTTCTCATTTCCAC
*IL-1β*	GCAACTGTTCCTGAACTCAACT	ATCTTTTGGGGTCCGTCAACT
*IL-10*	GCACTACCAAAGCCACAAAGC	GTCAGTAAGAGCAGGCA
*Β-actin*	CGGTTCCGATGCCCTGAGGCTCTT	CGTCACACTTCATGATGGAATTGA

**Table 3 nutrients-15-03029-t003:** Microbiome biomarker diversity.

Taxon Name	Taxon Rank	LDA Effect Size	N	NC	EHG
Proteobacteria	Phylum	5.05	1.27	24.36	6.22
Clostridia	Class	5.18	40.06	8.07	16.20
Bacilli	Class	5.11	6.11	33.34	23.43
Gammaproteobacteria	Class	5.07	0.03	24.10	5.41
Clostridiales	Order	5.18	40.06	8.07	16.20
Lactobacillales	Order	5.08	6.06	31.63	21.69
Enterobacterales	Order	5.07	0.03	24.08	5.35
Muribaculaceae	Family	5.13	33.74	6.68	17.51
Enterobacteriaceae	Family	5.07	0.03	24.02	5.34
Lactobacillaceae	Family	4.99	6.05	26.58	21.12
Bacteroidaceae	Family	4.98	11.41	24.69	31.33
*Escherichia*	Genus	5.07	0.03	22.78	5.16
*Lactobacillus*	Genus	0.03	22.78	22.78	21.12
*Escherichia coli* group	Species	5.07	0.03	22.78	5.15
*Lactobacillus murinus* group	Species	4.98	5.55	24.23	20.27

Treatment groups: N = normal control; NC = negative control (5% dextran sulfate sodium [DSS]); EHG = enzymatically hydrolyzed *Glycine max* + 5% DSS.

## Data Availability

The data presented in this study are available in this article.

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
