# Peer review of "Anti-Inflammatory Effect and Signaling Mechanism of Glycine max Hydrolyzed with Enzymes from Bacillus velezensis KMU01 in a Dextran-Sulfate-Sodium-Induced Colitis Mouse Model"

_nutrients, 2023, doi:10.3390/nu15133029_

Round 1

Reviewer 1 Report

This manuscript focuses on the obvious advantages of Glycine max hydrolyzed with enzymes from Bacillus velezensis KMU01 in preventing and treating colitis. The experimental setting is reasonable and rigorous and the data are properly supplemented.

Concerns

1In the animals of the materials and methods, the relevant information of mice was repeated; References are repeated in the references section.

2In 204 lines, please verify that if figure 1B-1D illustrates the results of reduction in colon length.

3In 235 and 239 lines, the scale bar is inconsistent with the values indicated in the picture.

4In 249 lines, figure legend does not describe the contents of the E F G H diagrams.

5Only the time of EHG pretreatment was introduced, but the time of treating cells with LPS was not described.

Reviewer 2 Report

The manuscript "Anti-inflammatory effect and signaling mechanism of Glycinemax hydrolyzed with enzymes from Bacillus velezensis KMU01 in a dextran sulfate sodium-induced colitis mouse model" was well written, in addition, the methods and statical analysis are consistent with the purpose. It presents an interesting topic and the results are robust and suitable for the conclusions.  However, minor points should be improved:

1. What is the number of animals used?

2. How many repetitions were performed in each experiment?

3. Normally, in in vivo assays, it is recommended that two or more concentrations be used, why was only one concentration of EHG used?

4. Why the references are duplicates? 

Minor editing of English language required. 

Round 2

Reviewer 1 Report

Accept